# The Effect of Sodium Reduction by Sea Salt and Dry Sourdough Addition on the Wheat Flour Dough Rheological Properties

**DOI:** 10.3390/foods9050610

**Published:** 2020-05-10

**Authors:** Andreea Voinea, Silviu-Gabriel Stroe, Georgiana Gabriela Codină

**Affiliations:** Faculty of Food Engineering, Stefan cel Mare University of Suceava, 720229 Suceava, Romania; andy_v93@yahoo.com (A.V.); codina@fia.usv.ro (G.G.C.)

**Keywords:** sea salt, dry sourdough, optimization, rheological properties

## Abstract

The aim of this research was to investigate a technological approach to decrease the sodium content from bakery products in order to respond to the World Health Organization (WHO)’s recommendation to reduce dietary salt intake. Due to the fact that sodium chloride is one of the main ingredients from baking products that affects dough rheology and therefore the technological process of the bakery products, it is important to evaluate these properties. This study analyzes the effect of sea salt with low sodium content (SS) and dry sourdough from wheat flour (SD) as substitutes for sodium chloride on dough rheological properties and on mixing, extension, pasting, and fermentation process by using Farinograph, Extensograph, Amylograph, Falling Number, and Rheofermentometer devices. The results were analyzed using response surface methodology. SS presented a strengthening effect on the gluten network whereas SD presented a weakening one. On extension properties, SS and SD presented a significant positive effect (*p* < 0.01) on resistance to extension (R_50_) and maximum resistance to extension (R_max_) values. For pasting properties, SS increased peak viscosity and falling number values whereas SD decreased them. On fermentation properties, SS decreased the maximum height of gaseous production and total CO_2_ volume production and increased the retention coefficient whereas SD presented an antagonistic effect on these parameters.

## 1. Introduction

The World Health Organization (WHO) promotes and encourages industries to reduce salt from foods in order to achieve a maximum salt consumption of 5 g per day for adults [1]. This recommendation is due to the fact that a high dietary sodium intake can lead to an increase in blood pressure with the occurrence of hypertension and chronic diseases such as cardiovascular ones [2,3]. Cardiovascular diseases are the leading cause of mortality in the world, in Europe alone it is estimated that this disease accounts every year for over 4 million deaths representing almost half of all its deaths [4]. Beside cardiovascular diseases, high sodium consumption can lead to other negative health problems such as obesity, osteoporosis, renal diseases, gastric cancer, etc. [5]. However it seems that no European country meets the WHO salt consumption recommendation level (≤5 g/day), and therefore nowadays a salt reduction strategy of 16% over 4 years is applied in order to achieve a sodium chloride population intake less than 5 g per day by 2025 [6,7]. Bakery products along with bread represent one of the main sources of sodium in the daily intake of the population, and therefore, many strategies to reduce salt intake have been undertaken to reformulate baked products [8,9]. There are many approaches to decrease sodium chloride in baked products while maintaining their high quality. The most common ones are sodium replacement with different substitutes such as KCl, calcium salts, magnesium salts, etc. [10,11,12] and intensification by the addition of different ingredients for the salty flavor perception [8,13,14,15]. In bakery products, sodium chloride plays an important role from the sensory point of view. Moreover, it has an important impact on the bread making technological process. In order to understand the impact of sodium chloride replacement on the bakery products’ technological process, its influence on dough processing has to be known. Studies show that sodium chloride has an important effect on gluten structure development, dough fermentation process, and water activity level in the bakery products [8]. Sodium chloride makes stronger wheat flour dough, with more stability and less weakening. A more stable dough structure contributes to gas cell formation and retention during the fermentation process [7,16]. Furthermore, a decrease in the level of salt addition in dough recipes influences yeast activity. In the absence of sodium chloride addition, the yeast activity is stimulated. This fact leads to an “over proved” dough and a significant increase of the dough height. The gas release is not highly retained by the dough system due to its weakening in the absence of salt. Bread products without salt addition will be in a flattened form with much larger pores due to the dough’s stickiness and cannot retain the gases formed which will diffuse or escape [17]. In addition, the bread will be pale in color, due to the intense yeast fermentation, which will consume large amounts of sugars. As a result, in the baking process, the wheat flour dough will present fewer amounts of sugars to form melanoidins to give color to the bread crust [18]. The aim of this study was to investigate sodium reduction in bakery products by using a sodium chloride substitute, i.e., sea salt with low sodium content, and an improver of bread flavor perception, i.e., dry sourdough. For this purpose, these ingredients namely sea salt and dry sourdough from wheat flour were used in different combinations in order to investigate their effect on dough rheological properties and to optimize a formulation of them by using response surface methodology (RSM). Although some studies were made on the effect of sea salt as a sodium substitute [19,20] or of sourdough type [13,21,22] on dough rheological properties, to our knowledge no study has been made on the combined effect of sea salt–dry sourdough formulation on dough rheological properties.

## 2. Materials and Methods

### 2.1. Materials

A refined wheat flour of 650 type (harvest 2019) provided by S.C. Mopan S.A. Company (Suceava, Romania) was used. The low-sodium sea salt is a natural mineral salt of the Dead Sea provided by BK Giulini Corp (New York, NY, United States) being a natural hydrated potassium magnesium chloride. Its mineral structure includes magnesium as magnesium chloride (31–35%), potassium as potassium chloride (21–27%), and sodium as sodium chloride (max. 7%). The dry sourdough based on wheat flour was provided by Enzymes & Derivates S.A. Company (Neamt, Romania). Sodium chloride was purchased from the Romanian market. The wheat flour analyzed through Romanian and international methods presented the following values: 0.65 g/100 g ash content International Association for Cereal Chemistry (ICC 104/1), 12.67 g/100 g protein content (ICC 105/2), 30 g/100 g wet gluten content (ICC 106/1), 6 mm gluten deformation index, 14 g/100 g moisture content (ICC 110/1), and 322 s Falling Number value (ICC 107/1).

### 2.2. Rheological Properties of Dough during Mixing and Extension

Water absorption (WA), dough development time (DDT), dough stability (ST), and degree of softening at 10 min (DS) were analyzed using the Farinograph device (Brabender, Duisburg, Germany, 300 g capacity) following ICC method 115/1. The dough prepared based on Farinograph data was analyzed in the Extensograph device (Brabender, Duisburg, Germany) following ICC method 114/1. The measured parameters analyzed through the Extensograph were as follows: energy (E), resistance to extension (R_50_), extensibility (Ext), maximum resistance to extension (R_max_), and ratio number (R/E) at a proving time of 135. 

### 2.3. Pasting Properties

The pasting properties of flour mixes were measured by an Amylograph device (Brabender OGH, Duisburg, Germany) following ICC method 126/1 and by Falling Number device following ICC method 107/1. The measured parameters analyzed through Amylograph and Falling Number were as follows: temperature at peak viscosity (T_max_), gelatinization temperature (T_g_), temperature at peak viscosity (T_max_), and falling number index value (FN).

### 2.4. Rheological Properties of Dough during Fermentation

The rheological properties of dough during fermentation were measured by Rheofermentometer (Chopin Rheo, type F3, Villeneuve-La-Garenne Cedex, France) following the American Association of Cereal Chemists (AACC) method 89-01.01. The measured parameters analyzed through Rheofermentometer were as follows: maximum height of gaseous production (H’_m_), total CO_2_ volume production (VT), volume of the gas retained in the dough at the end of the test (VR), and retention coefficient (CR).

### 2.5. Experimental Design and Statistical Analysis

Response surface methodology (RSM) was used to study the simultaneous effects of Dead Sea salt (SS) and dry sourdough (SD) on the wheat flour dough rheological properties. An important advantage of the RSM methodology is that a reduced number of experimental runs is necessary to provide information on statistical results [23]. The design of experiment (DOE) used in this study was Design Expert software version 12, which includes ANOVA statistical models (trial version, Stat-Ease, Inc., Minneapolis, MN, USA). The experimental plan, with the coded and real values, comprising 13 experiences is shown in Table 1.

In this study, two independent variables were chosen for the statistical experiment design as follows: the influence of the variation of the sea salt amount (*A = X*_1_) and dry sourdough amount (*B = X*_2_) on the rheological properties of wheat flour dough. The amounts of sea salt used in the experiment were between 0.3 and 1.5 g/100 g wheat flour, and the amounts of dry sourdough used were 0.5–5 g/100 g wheat flour. In order to eliminate the measurement errors, the values of the rheological parameters on Farinograph, Extensograph, and Falling Number were carried out in duplicate in the statistical processing using the average values obtained. For a complete study of the dough’s rheological behavior different test methods were used in order to analyze the effect of SS and SD on mixing, extension, pasting, and fermentation properties. The rheological parameters (dependent variables) obtained through the Farinograph were as follows: WA—water absorption (Y1); DT—development time (Y2); ST—stability of dough (Y3); and DS—degree of softening (Y4). The rheological parameters obtained through the Extensograph (dependent variables) were as follows: E—energy (Y5); R50—resistance to extension up to 50 mm (Y6); Ext—extensibility (Y7); R_max_—maximum resistance (Y8). The falling number index values (Y9) and the Amylograph parameters were also determined: Tg—gelatinization temperature (Y10); PV_max_—peak viscosity (Y11), T_max_—temperature at peak viscosity (Y12), H’_m_—height under constraint of dough at maximum development time (Y13), VT—total volume of CO_2_ produced during fermentation (Y14), VR—volume of the gas retained in the dough at the end of the test (Y15); and CR—retention coefficient (Y16). The quadratic models to describe the process mathematically were developed using the general expression of a polynomial equation coded by the second degree as follows:(1)$$Y=f\left(X_{i},\;X_{j}\right)=\beta_{0}+\overset{n}{\underset{i=1}{\sum }}\beta_{i}\cdot X_{i}+\overset{n-1}{\underset{i=1}{\sum }}\overset{n}{\underset{j=i+1}{\sum }}\beta_{ij}\cdot X_{i}\cdot X_{j}+\overset{n}{\underset{i=1}{\sum }}\beta_{ij}\cdot X_{i}^{2}+\varepsilon \;,$$
where *Y*—predicted response; $\beta_{0}$—interception coefficient; $\beta_{i}$ and $\beta_{j}$—coefficients of the linear effects; $\beta_{ii}$ and $\beta_{jj}$—coefficients of the quadratic effects, and $\beta_{ij}$—coefficients of the interaction effects for the *X_i_* and *X_j_* independent variables; and ε is the random error. A coding of process variables was used in order to simplify the calculation procedures. The non-significant terms from the two-coded polynomial equation to simplify it were excluded. The adequacy of the obtained model was evaluated using the *F* ratio and the coefficient of determination (*R*^2^). The significant model terms were evaluated by the *p* value at 95% confidence level. The three-dimensional graphical representation of the response surfaces was made as a function of the independent variables.

For sodium chloride, data were expressed as means ± standard deviations. The determinations were made in triplicate. Statistical analysis was performed using Microsoft Excel 2013. Differences were considered to be significant at a validity of α = 0.95.

## 3. Results and Discussion

### 3.1. Fitting Models

The most-fitting models (quadratic) were obtained for the Farinograph parameters water absorption (WA), dough development time (DT), dough stability (ST), degree of softening at 10 min (DS); for Amylograph parameters peak viscosity (PV_max_) and temperature at peak viscosity (T_max_); and for the Rheofermentometer value retention coefficient (CR). For the Extensograph parameters, resistance to extension up to 50 mm (R_50_) and maximum resistance (R_max_), the models were 2FI. Additionally, a 2FI model was obtained for falling number (FN) values. Linear models were obtained for the Extensograph parameters energy (E) and extensibility (Ext), as well as for the Rheofermentometer parameter volume of the gas retained in the dough at the end of the test (VR).

### 3.2. Mixing and Extension Values for the Formulation Samples

The ANOVA results applied on mixing and extension values for the formulation samples indicated that sea salt presented a highly significant effect (*p* < 0.01) on WA, DS, E, R_50_, and R_max_ whereas SD affected WA, ST, DS, and Ext. Increasing the level of SD addition led to an increase of Tg, T_max_, CR, E, R_50_, and R_max_ values. Similar results during mixing were obtained by McCann and Day [24], Beck et al. [25,26], Utkameran et al. [27], Miller and Jeong [19], and Nogueira and Kussano [13] and during extension by Miller and Jeong [19], Utkameran et al. [27], Lynch et al. [17], Nogueira and Kussano [13]. For both variables used it seems that SD presented a statistical effect on all the mixing and extension values (*p* < 0.1) whereas SS did not present any statistical effect on dough stability and extensibility values. It can be seen that SS decreased the VT and VR values. In Table 2, it can be observed that the obtained values of the adjusted determination coefficients are higher than 98% (Adjusted *R*^2^ > 0.9816) for the dependent variables WA, ST, and DS, which means that these equations are the most significant ones (*p* < 0.0001).

Furthermore, values higher than 90% (Adjusted *R*^2^ > 0.9038) of the adjusted determination coefficient were obtained for the dependence variables E, T_max_, and CR respectively. For PV_max_, VR, DT, and T_g_ response, the Dead Sea salt had a significant effect (*p* < 0.05).

The interaction between SS and SD on WA value is shown in Figure 1a. As it may be seen, both independent variables, singly and in combination, presented a highly significant effect (*p* < 0.01) on WA.

It may be noticed that WA value is decreased by SS addition and increased when SD is incorporated in dough recipe. These data are in agreement with those obtained by Jekle et al. [5], which concluded that any chloride salt decreased water absorption of wheat flour. Salt, such as sea salt, due to its ionic nature, interacts with macromolecules and water from the dough system changing the WA of wheat flour. Due to higher hydrophobic interactions, the gluten proteins interact to a higher extent leading to a reduced water uptake ability. The decreased WA value with the increased level of salt addition confirms the results from many studies previously made [12,24,25,26,27,28]. Contrarily, SD presented a positive effect on WA, this result is in agreement with those reported by Nogueira and Kussano [13].

From Figure 1b, it may be shown that both independent variables have a statistically significant effect (*p* < 0.01) on degree of softening (DS). Sea salt presents a negative effect on DS whereas dry sourdough presents a positive one. This fact indicates that dough becomes stronger when SS is incorporated and softer when SD is added in wheat flour. A strengthening effect of dough by SS addition has been previously reported by [29,30]. This may be attributed to the closer molecular association between gluten proteins due to higher hydrophobic interactions between these proteins. According to Wellner et al. [30], this close association is related to the high content of about 40% hydrophobic amino acids from gluten proteins in the structure. In general, the strengthening effect of wheat flour dough has been reported for all chloride salts that have been incorporated in dough system. Bernklau et al. [29] reported a higher strengthening effect when KCl was used even with just a partial substitution of NaCl. Moreover, it is well known that the magnesium ions cause a hydrated and soluble effect on gluten proteins due to the fact that they are situated in the destabilizing zone of the Hofmeister series with a strong effect on hydrophobic interactions between gluten proteins, which will promote their aggregation [31,32]. Due to the fact that sea salt is a mix of magnesium chloride, potassium chloride, and sodium chloride, its strengthening effect on the dough system is explainable.

The graphical representations of the Extensograph parameters curve resistance to extension (R_50_) and maximum resistance to extension (R_max_) in relation to the level of the SS and SD addition are shown in Figure 1c,d. In these cases, all the model terms present a significant positive effect (*p* < 0.01) on R_50_ and R_max_ values. According to Ortolan et al. [33], when salt was used in the dough recipe for the extensographic analysis, it favored the hydrophobic interactions between gluten proteins, a fact that will strengthen the gluten network as a result of a closer alignment between molecules. The increase of the R_50_ and R_max_ values indicates an increase of dough hardening with the increasing level of SS and SD addition. Similar results were reported by Miller and Jeong [19] when sea salt was added in wheat flour and by Nogueira et al. [13] when sodium chloride and SD were incorporated in the dough recipe.

### 3.3. Pasting Parameters Values of the Formulation Samples

The effects of the independent variables, SS and SD, on pasting properties of formulation samples obtained through Falling Number and Amylograph device are shown in Table 3 and Figure 2.

SD presented a significant effect (*p* < 0.1) on all viscometric parameters, whereas SS presented a positive significant effect (*p* < 0.05) on peak viscosity (PV_max_) and temperature at peak viscosity (T_max_). From all the predictive models obtained for pasting parameters of the formulation sample, the most significant one (*p* < 0.0001) was that for T_max_ value, which presented the highest determination coefficient (*R*^2^ = 0.97) value. The gelatinization temperature (T_g_) decreased when the SS and SD addition decreased, indicating that in the presence of SD and SS starch gelatinization is delayed. The addition of SS led to an increase of PV_max_ and FN values. Nogueira et al. [13] consider that this increase of the peak viscosity by salt addition may be attributed to the fact that the enzymatic activity in the dough system is suppressed in the presence of salt. Another explanation may be the fact that the presence of salt may cause the starch granules to remain intact for a long period of time before fragmentation. The falling number value is also an expression of mix viscosity, an increase of this value leading to an increase of FN value too. Regarding the SD influence on PV_max_ and FN values, it presents a negative effect on PV_max_ and FN values. Because FN value decreased with SD addition, it may be concluded that SD increased α amylase activity of the mix flours. This behavior may be due to a higher α amylase activity from the SD system, taking into account that this ingredient is the result of a fermentation process of wheat flour and it possibly presents higher α amylase activity than that of the wheat flour used in this study. A higher α amylase activity decreased the viscosity of the mix flours and, therefore, decreased the peak viscosity value

### 3.4. Fermentation Parameters Values of the Formulation Samples

The effects of independent variables on fermentation parameter values obtained from the Rheofermentometer device are shown in Table 3. As it may be seen, SS presented a highly significant effect on all fermentation parameters (*p* < 0.01), whereas SD had a significant effect on maximum height of gaseous production (H’_m_), total CO_2_ volume production (VT), and retention coefficient (CR) of samples. Except on CR parameter value, SS presented a negative effect on all fermentation values, whereas SD presented a positive one. The positive or negative effects of independent variables on fermentation values are somehow interconnected. For example, a decrease of total CO_2_ volume production (VT) leads to a decrease of H’_m_. The lowering of the maximum height of gaseous production is a consequence of the fact that the CO_2_ produced by the yeast during fermentation is in a low amount and vice versa. Remarkably, the CR value, which is a measurement of the capability of dough to retain gases, is the only fermentation parameter found to be higher when SS is added in wheat flour. These results can be explained taking into account the strengthening effect of SS on the gluten network. Therefore, even if less CO_2_ is produced in doughs with higher salt levels, the strengthened gluten network can hold the gases formed, leading to higher amount of CO_2_ remaining in the dough and higher CR values [34].

The graphical representation of H’_m_ in relation with the level of SS and SD addition is shown in Figure 3a. Higher sea salt levels caused a decreased of H’_m_ whereas higher SD levels caused an increase. According to Jekle et al. [5], the addition of any kind of chloride salt leads to a decrease of maximum dough height due to it direct influence on yeast metabolism by its osmotic pressure. The chloride salt inhibits yeast growth causing a decrease in yeast activity, leading to lower CO_2_ production and, therefore, to lower H’_m_ values. Similar effects on H’_m_ value by salt addition have also been reported by [17,19,24,25,34]. With regard to the effect of SD on H’_m_ value, these data are in agreement with those obtained by Clarke et al. [35], which concluded that an increase of the fermentation time will allow a higher expansion of the gluten network because of it softer nature induced by SD addition.

From the plot in Figure 3b, it may be seen that SS has a negative effect on total CO_2_ volume production (VT), whereas the SD has a positive one. This means that SS decreases the VT whereas SD increases it. This behavior is a consequence of the salt’s inhibiting effect on yeast, which limited gas production. According to Pasqualone et al. [34], due to its inhibitory effect, the fermentation time must be adapted when the salt level is changed. Without salt yeast ferments excessively in wheat dough, which may lead to bakery products of a poor quality. With the increasing level of salt addition in wheat flour, the leavening ability is decreased. Therefore, in the bread making process, the inhibiting effect of salt on yeast is used to control its activity. Regarding the positive effect of SD on VT, different studies show that in the presence of sourdough yeast fermentation is slower and more carbon dioxide is produced [36]. During the proofing process, CO_2_ is produced by the fermentation of carbohydrates by yeast. SD is obtained by wheat fermentation. Therefore, it may present a higher amylolytic activity, which hydrolyses starch leading to a higher amount of fermentable sugars for yeast. Moreover, in the case of SD addition, CO_2_ is produced by both yeast and lactic acid bacteria, which may led to a VT increase.

The quadratic model for the retention coefficient (CR) obtained is a significant one (*p* < 0.0001) with a negative effect provided by SD and a positive one provided by SS and interactions between SS and SD. The effects of SS and SD on CR were highly significant (*p* < 0.01). The effect of interactions between SS and SD is shown in Figure 3c. The contour plot indicates that with an increase in the level of SS the CR increased and with an increase in the level of SD the CR decreased. Due to its ionic nature, chloride salts positively influence the interactions between gluten strands in the wheat flour dough [25]. Therefore, an improvement of the gluten network takes place with the increased level of SS addition. Salt presents a strengthening effect on the gluten network, which makes the dough more capable of retaining the gas released by fermentation. Therefore, even if SS addition leads to lower VR and VT values, it increases the CR (VR/VT) value [28]. Contrary to the effect of SS, SD decreased the CR values probably due to the fact that through SD addition the dough becomes more acidic as the lactic acid bacterium from SD become dominant [35]. The pH decreased along with a higher proteolytic activity from the SD system and the dough may become weaker and less capable of retaining the gas released by fermentation [36].

### 3.5. Optimization of Sea Salt and Dry Sourdough Formulation

In order to calculate the optimal levels of the rheological parameters of the dough, Derringer’s desirability function was used (Equation (2)). Derringer’s desirability function allows finding experimental conditions (factor levels) to achieve, simultaneously, the optimal value for all evaluated variables [23,37].
(2)$$D={\left(d_{1}^{r_{1}}\cdot d_{2}^{r_{2}}\cdot \ldots \cdot d_{n}^{r_{n}}\right)}^{\frac{1}{\sum r_{i}}}$$
where d_1_, d_2_, …, d_n_ are the desirability indices for each dependent variable (d = 0 is least desirable and d = 1 is most desirable) and r_i_ is the relative importance of each dependent variable. The numerical simultaneous optimization in Design Expert software was done keeping the values of the independent variables, sea salt (SS) and dry sourdough (SD), in their range.

Applying the desirability function methodology, the optimal levels of the independent variables were obtained. A total of 10 solutions were generated after numerical optimization. Figure 4 shows the graphical representation of the desirability function or the chosen Solution 1 out of 10.

Thus, the optimum amounts are 1.396 g SS/100 g wheat flour and 2.683 g SD/100 g wheat flour. For these optimal solutions, WA—60.011%, DT—1.597 min, ST—1.332 min, DS—87.988 UB, T_g_—62.800 °C, PV_max_—1139.828 BU, T_max_—89.417 OC, FN—343.787 s, H’_m_—68.036 mm, VT—1322.238 mL, VR—1157.821 mL, CR—87.938%, E—83.139 cm^2^, R_50_—446.108 BU, Ext—123.014 mm, R_max_—300 BU, and R/E—3.727 parameters were obtained with a desirability function score of 0.447081.

### 3.6. Sodium Chloride Reduction from Bakery Products by Sea Salt and Dry Sourdough Optimum Formulation

According to the European Commission (EC) no. 1924/2006 regulation on “nutrition and health claims made on foods”, food products can claim on the label some aspects regarding the sodium/sodium chloride content as follows: reduced sodium/sodium chloride if it content is at least 30% lower than that of a similar product; low sodium/sodium chloride if the food products contains no more than 0.12 g of sodium/0.3 g sodium chloride per 100 g or per 100 mL; very low sodium/sodium chloride if the food product contains no more than 0.04 g of sodium/0.1 g sodium chloride per 100 g or per 100 mL; sodium-free or sodium chloride-free if the food products contains no more than 0.005 g of sodium/0.013 g sodium chloride per 100 g or per 100 mL. Our data indicate an optimum amount per 100 g wheat flour of SS and SD established through response surface methodology of 1.396 and 2.683 g, respectively. The SS ingredient is the only one that contains sodium to a level of a maximum of 7% as sodium chloride. Therefore the bakery products obtained through this recipe will present a maximum of 0.04 g of sodium/0.1 g sodium chloride per 100 g, which will classify them as very low sodium/sodium chloride products. Nowadays, there is a wide variation regarding the level of sodium/sodium chloride in bakery products. For example, there may be a difference of 1 g of sodium chloride between some bakery products from Germany (highest) and UK (lowest) [7]. In general, the level of sodium chloride may be around the values of 1.00–1.50 g/100 g in bread and bakery products. Of course, the values can be lower or higher depending on country or bakery type. Although bakery products do not contain such high levels of sodium chloride, they make an important contribution to daily sodium intake due to their high consumption by the population. Taking that into account our optimum recipe obtained through response surface methodology, this formulation leads to bakery products with a sodium level 10–15 times lower than that of regular ones found today on the market. To our knowledge, no further studies have proposed this combination as a sodium chloride replacer in order to obtain bakery products with low sodium content.

### 3.7. Effect of Sodium Chloride Addition on Dough Rheological Properties Compared to the Optimum Amounts of SS and SD Obtained through Response Surface Methodology

For bakery products, added salt has essential functions (on dough rheology and, therefore, dough technological behavior, and bakery product quality such as flavor, texture, and shelf-life) and, therefore, the effects of reducing the NaCl must be carefully considered. The sodium chloride technological effect consists especially in its influence on dough rheological properties. According to the mixing data shown in Table 4, NaCl addition decreases WA value and increases the DT and ST values.

This behavior is due to the fact that the sodium chloride is primarily related to the change of gluten proteins’ hydration, which modifies the ratio between free and bound water. This is attributed to the conformational changes of the gluten proteins in the presence of sodium chloride. The conformational changes would take place due to the interaction of the sodium chloride ions with the electrically charged groups of the gluten proteins. As a result, the intermolecular and intramolecular electrostatic repulsion forces are reduced, and the protein molecule becomes more compact. The result of these conformational changes may lead to a WA value decrease and also to a DT and ST increase. The increase of the DT values shows that the NaCl delays the formation of gluten during mixing, thus increasing the mixing time. A strengthening effect of NaCl also leads to a higher dough resistance as can be seen from the increased Extensograph value for R_50_ and R_max_. In literature, similar data on the effect of NaCl on dough rheological properties during mixing have also been described [24,25,27,29]. From the Amylograph data shown in Table 5, the addition of NaCl delays the starch gelatinization process as can be seen by the increase of T_g_, this result is in agreement with those obtained by [38]. Furthermore, an increase of the FN, PV_max_ and T_max_ values with the increase level of NaCl addition may be noticed. At low salt levels, with the addition of 0.3–0.6 g/100 g, yeast activity is stimulated. This fact is attributed to the inhibition of the toxic effect by salt of the thionine (protamine), which inhibits the yeast fermentation activity. On higher levels of NaCl addition, the fermentative process is inhibited. The decrease of the multiplication and fermentative activity of the yeast on salt additions levels higher than 0.6 g/100 g may be attributed to the plasmolysis action of the yeast cells. At higher levels of sodium chloride addition, the yeast activity decreased due to the osmotic pressure and the action of salt ions on the semi-permeable membrane of yeast cells [12].

Our proposed strategy to reduce the sodium from bakery products is to replace the sodium chloride with SS and SD addition in wheat flour. In general, the SS addition has a similar effect on dough rheological properties as sodium chloride salt. However, SD presented a contrary effect than SS and of course sodium chloride addition in wheat flour on WA, ST, DS, E, Ext, FN, and all Rheofermentometer values H’_m_, VT, VR, and CR. Both independent variables presented a similar effect to sodium chloride on R_50_, R_max_, T_g_, PV_max_. This fact will lead to a more or less significant effect of SS and SD addition in wheat flour than that of sodium chloride on dough rheological properties values. The effect will be higher if both independent variables have the same impact (positive or negative) on dough rheological properties and lower if SS and SD have a contrary effect on rheological values. The optimum formulation obtained by us through response surface methodologies was 1.396 g SS/100 g and 2.683 g SD/100 g addition in wheat flour. For this formulation, we obtained weaker dough than those with sodium chloride addition according to the Farinograph data. According to Extensograph data, we obtained dough with extensibility similar to those with 0.6% sodium chloride addition and R_max_ and E similar to those with 1.5% NaCl addition. The FN values were similar to those of wheat flour with 0.9% sodium chloride addition and T_g_, PV_max_, T_max_ were similar with wheat flour samples with 0.3%, 0.6%, and 0.9% sodium chloride addition. In general, the Rheofermentometer values for the optimum formulation were similar to those for the dough sample with 0.9% sodium chloride addition in wheat flour.

## 4. Conclusions

The study presented in this paper showed that, using the response surface methodology (RSM), it is possible to optimize the amounts of sea salt and dry sourdough in order to obtain the best values for the rheological parameters of the wheat flour dough. The mathematical models obtained and used for the different response variables’ analysis showed values of Adjusted *R*^2^ > 0.7 (except for DT, FN, T_g_, and VR) and *p* < 0.01. In general, the independent variables used presented a contrary effect. During mixing and extension, SS strengthens the wheat flour dough whereas SD weakness it. On pasting properties, SS increased peak viscosity and falling number values whereas SD decreased them. According to Rheofermentometer characteristics, SS presented a negative effect on all fermentation values whereas SD presented a positive one except on the CR parameter value. The optimum levels obtained for the independent variables by the numerical optimization method were 1.396 g SS/100 g wheat flour and 2.683 g SD/100 g wheat flour.

## Figures and Tables

**Figure 1 foods-09-00610-f001:**
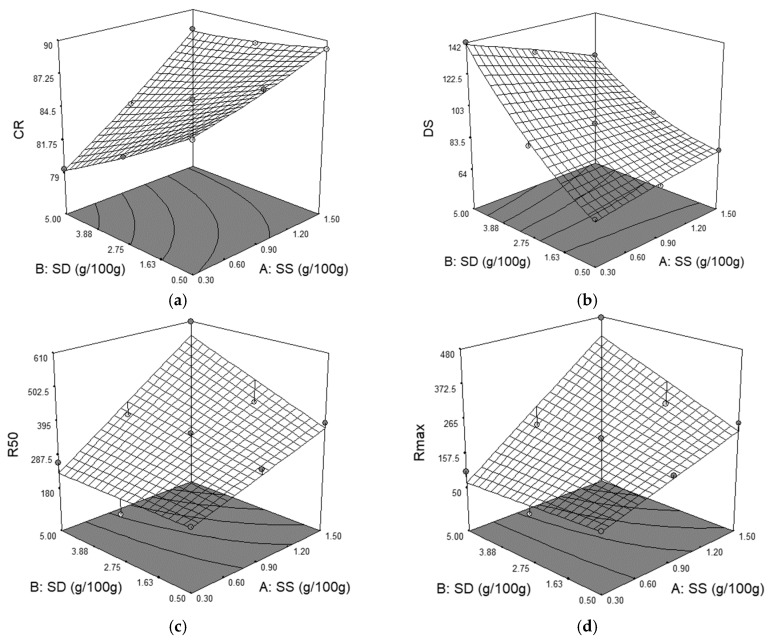
The graphical representations of the Farinograph and Extensograph parameters: (**a**) water absorption (WA); (**b**) degree of softening at 10 min (DS); (**c**) resistance to extension up to 50 mm (R_50_); (**d**) maximum resistance to extension (R_max_).

**Figure 2 foods-09-00610-f002:**
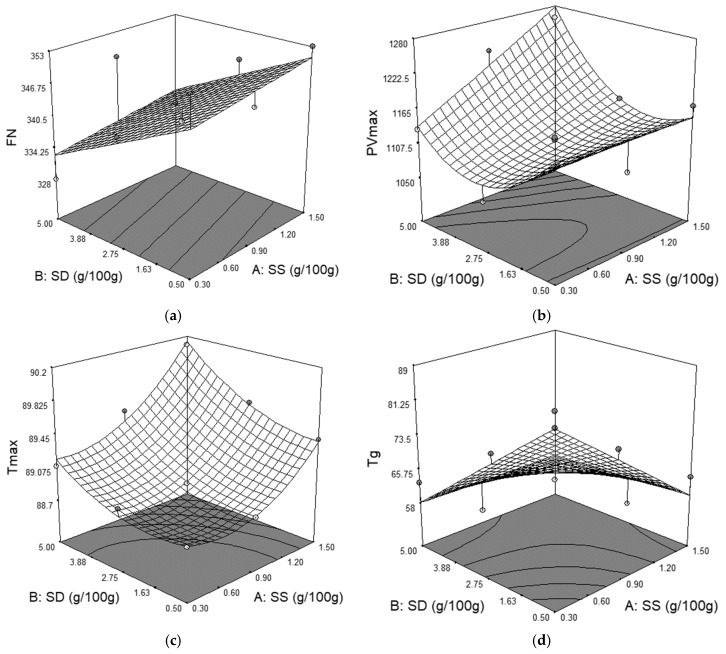
The graphical representations of the Falling Number and Amylograph parameters: (**a**) falling number value (FN); (**b**) peak viscosity (PV_max_); (**c**) temperature at peak viscosity (T_max_); and (**d**) gelatinization temperature (T_g_).

**Figure 3 foods-09-00610-f003:**
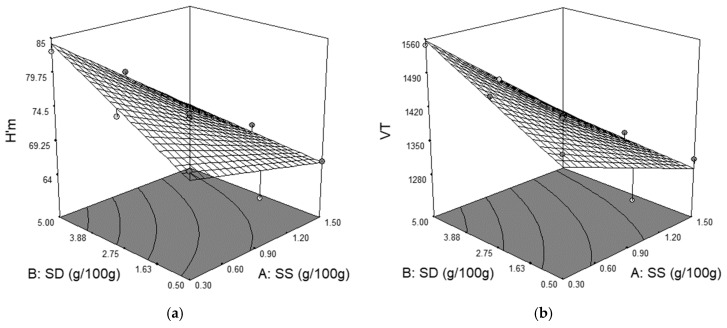
The graphical representations of the Rheofermentometer parameters: (**a**) maximum height of gaseous production (H’_m_); (**b**) total CO_2_ volume production (VT); and (**c**) retention coefficient (CR).

**Figure 4 foods-09-00610-f004:**
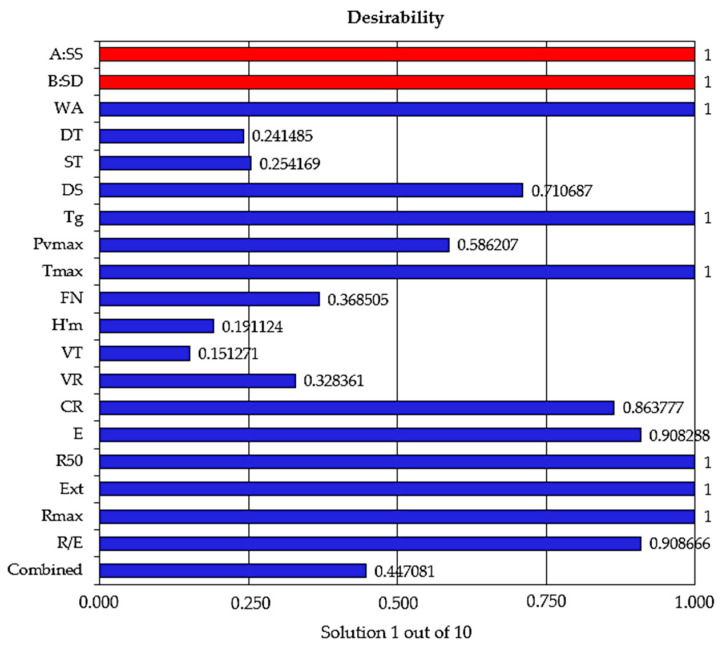
Desirability function scores for the independent and dependent variables.

**Table 1 foods-09-00610-t001:** Coded and real values of formulation factors used in the experimental design.

Run	Coded Value		Real Value	
	X_1_	X_2_	SS (g/100 g)	SD (mg/100 g)
1	0	0	0.9	2.75
2	0	0	0.9	2.75
3	0	0	0.9	2.75
4	0	0	0.9	2.75
5	0	1	0.9	5.00
6	1	1	1.5	5.00
7	1	−1	1.5	0.50
8	−1	0	0.3	2.75
9	0	−1	0.9	0.50
10	0	0	0.9	2.75
11	−1	1	0.3	5.00
12	1	0	1.5	2.75
13	−1	−1	0.3	0.50

SS—sea salt; SD—dry sourdough.

**Table 2 foods-09-00610-t002:** Effects of formulation factors expressed as their corresponding coefficients on the predictive models for dough rheological properties during mixing of sea salt–dry sourdough mixtures.

Factors	Parameters						
	Farinograph			Extensograph (Proving Time 135 min)			
	WA (%)	ST (min)	DS (UB)	E (cm^2^)	R_50_ (BU)	Ext (mm)	R_max_ (BU)
Constant	60.49	1.27	92.31	66.00	350.08	122.69	201.08
*A*	−0.85 ***	0.0000	−4.00 ***	20.50 ***	119.17 ***	−0.1667	122.63 ***
*B*	1.45 ***	−0.7333 ***	28.50 ***	−6.00 **	46.17 **	15.33 ***	40.18 **
*A × B*	0.05	−0.0250	−9.50 ***	-	47.25 **	-	53.53 **
*A* ^2^	0.3983 ***	0.0586	−0.5862	-	-	-	-
*B* ^2^	−0.3017 ***	0.4586 ***	6.91 ***	-	-	-	-
Adjusted *R*^2^	0.9964	0.9816	0.9937	0.9038	0.8407	0.7443	0.8340
*p*-value	<0.0001 ***	<0.0001 ***	<0.0001 ***	<0.0001 ***	0.0002 ***	0.0004 ***	0.0002 ***

Significant at *p* < 0.01 ***, at *p* < 0.05 **, at *p* < 0.1 *. A—sea salt (g/100 g); B—dry sourdough (g/100 g); Adjusted *R*^2^ is a measure of the fit of the model. WA—water absorption; ST—dough stability; DS—degree of softening; E—energy; R_50_—resistance to extension up to 50 mm; Ext—extensibility; R_max_—maximum resistance.

**Table 3 foods-09-00610-t003:** Effects of formulation factors expressed as their corresponding coefficients on the predictive models for dough rheological properties during fermentation, gelatinization properties, and α-amylase activity of sea salt–dry sourdough mixtures.

Factors	Parameters				
	PV_max_ (BU)	T_max_ (OC)	H’_m_ (mm)	VT (mL)	CR (%)
Constant	1112.55	88.92	72.47	1394.00	85.19
*A*	32.83 **	0.3833 ***	−5.30 ***	−86.00 ***	3.10 ***
*B*	25.50 *	0.2833 ***	3.80 ***	46.17 ***	−2.00 ***
*A* × *B*	38.00 **	0.1000 **	−2.62 **	−31.00 **	1.35 ***
*A* ^2^	2.57	0.2776 ***	-	-	0.2241
*B* ^2^	67.57 ***	0.1776 ***	-	-	0.2241
Adjusted *R*^2^	0.7258	0.9711	0.8514	0.8833	0.9942
*p*-value	0.0104 **	<0.0001 ***	<0.0001 ***	<0.0001 ***	<0.0001 ***

Significant at *p* < 0.01 ***, at *p* < 0.05 **, at *p* < 0.1 *. A—sea salt (g/100 g); B—dry sourdough (g/100 g); Adjusted *R*^2^ is a measure of the fit of the model. PV_max_—peak viscosity, T_max_—temperature at peak viscosity, H’_m_—height under constraint of dough at maximum development time, VT—total volume of CO_2_ produced during fermentation, CR—retention coefficient.

**Table 4 foods-09-00610-t004:** Effects of different levels of sodium chloride addition on dough rheological properties during mixing.

NaCl Level (g/100 g)	Parameters							
	Farinograph				Extensograph (Proving Time 135 min)			
	WA (%)	DT (min)	ST (min)	DS (UB)	E (cm^2^)	R_50_ (BU)	Ext (mm)	R_max_ (BU)
0.0	60.5 ± 0.2 ^a^	1.9 ± 0.1 ^a^	2.0 ± 0.1 ^a^	76 ± 3.2 ^a^	57 ± 0.9 ^ab^	327 ± 2.3 ^ab^	115 ± 1.3 ^a^	226 ± 2.1 ^ab^
0.3	59.9 ± 0.1 ^a^	2.0 ± 0.1 ^a^	2.2 ± 0.1 ^a^	72 ± 2.1 ^a^	61 ± 1.2 ^ab^	331 ± 1.6 ^ab^	119 ± 1.4 ^a^	230 ± 2.9 ^ab^
0.6	58.7 ± 0.2 ^a^	2.1 ± 0.1 ^a^	2.4 ± 0.1 ^a^	69 ± 1.4 ^a^	66 ± 1.6 ^ab^	347 ± 3.9 ^ab^	124 ± 1.2 ^a^	238 ± 2.2 ^ab^
0.9	58.2 ± 0.1 ^a^	2.2 ± 0.1 ^a^	2.7 ± 0.2 ^a^	66 ± 1.5 ^a^	75 ± 2.1 ^ab^	371 ± 6.2 ^ab^	135 ± 1.6 ^a^	254 ± 3.1 ^ab^
1.2	57.6 ± 0.1 ^a^	2.3 ± 0.1 ^a^	3.2 ± 0.2 ^a^	64 ± 2.2 ^a^	81 ± 1.7 ^a^	395 ± 5.6 ^a^	126 ± 1.4 ^a^	277 ± 4.3 ^a^
1.5	57.2 ± 0.1 ^b^	2.7 ± 0.2 ^a^	3.6 ± 0.2 ^a^	61 ± 1.3 ^b^	85 ± 2.2 ^b^	416 ± 3.9 ^b^	122 ± 1.1 ^b^	298 ± 5.6 ^b^

The values are means ± standard deviations of three replicates. Means in the same column followed by different superscript letters (a,b) are significantly different at *p* < 0.05. Identical superscript letters (a,b) indicate no significant difference at *p* < 0.05. WA—water absorption; DT—development time; ST—dough stability; DS—degree of softening; E—energy; R_50_—resistance to extension up to 50 mm; Ext—extensibility; R_max_—maximum resistance.

**Table 5 foods-09-00610-t005:** Effects of different levels of sodium chloride addition on dough rheological properties during fermentation, gelatinization properties, and α-amylase activity.

NaCl Level (g/100 g)	Parameters							
	FN (s)	Tg (°C)	PV_max_ (BU)	T_max_ (°C)	H’_m_ (mm)	VT (mL)	VR (mL)	CR (%)
0.0	322 ± 2.0 ^ab^	62.3 ± 0.12 ^a^	1081 ± 4.7 ^ab^	88.4 ± 0.18 ^a^	67.3 ± 1.8 ^a^	1287 ± 8.3 ^a^	1066 ± 7.3 ^a^	82.8 ± 1.3 ^ab^
0.3	331 ± 0.9 ^a^	62.7 ± 0.09 ^a^	1097 ± 2.2 ^ab^	88.9 ± 0.22 ^a^	73.2 ± 1.4 ^a^	1429 ± 6.9 ^a^	1130 ± 5.4 ^a^	79.0 ± 0.9 ^ab^
0.6	338 ± 1.3 ^a^	63.2 ± 0.10 ^a^	1141 ± 3.9 ^ab^	89.2 ± 0.32 ^a^	77.7 ± 0.9 ^a^	1512 ± 7.9 ^a^	1166 ± 6.3 ^a^	77.1 ± 1.0 ^ab^
0.9	347 ± 1.2 ^a^	63.6 ± 0.13 ^a^	1179 ± 5.9 ^ab^	89.5 ± 0.22 ^a^	66.3 ± 1.3 ^a^	1227 ± 6.7 ^a^	1045 ± 7.4 ^a^	85.2 ± 1.4 ^a^
1.2	352 ± 0.4 ^a^	63.8 ± 0.07 ^a^	1267 ± 6.8 ^a^	89.7 ± 0.38 ^a^	61.6 ± 1.7 ^a^	1114 ± 9.8 ^a^	970 ± 6.7 ^a^	87.1 ± 1.6 ^a^
1.5	358 ± 1.6 ^b^	63.9 ± 0.10 ^b^	1290 ± 3.1 ^b^	89.9 ± 0.24 ^b^	50.6 ± 1.8 ^b^	939 ± 3.5 ^b^	855 ± 2.3 ^b^	91.0 ± 1.1 ^b^

The values are means ± standard deviations of three replicates. Means in the same column followed by different superscript letters (a,b) are significantly different at *p* < 0.05. Identical superscript letters (a,b) indicate no significant difference at *p* < 0.05. FN—falling number; Tg—gelatinization temperature; PV_max_—peak viscosity, T_max_—temperature at peak viscosity, H’_m_—height under constraint of dough at maximum development time, VT—total volume of CO_2_ produced during fermentation, VR—volume of the gas retained in the dough at the end of the test, CR—retention coefficient.

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
