# Peer review of "The Effect of Sodium Reduction by Sea Salt and Dry Sourdough Addition on the Wheat Flour Dough Rheological Properties"

_foods, 2020, doi:10.3390/foods9050610_

Round 1

Reviewer 1 Report

Manuscript ID: foods-786800

Title: The effect of sodium reduction by sea salt and dry sourdough addition on the wheat flour dough rheological properties.

The effect of sea salt and dry sourdough from wheat flour as substitutes for sodium chloride on dough rheological properties on mixing, extension, pasting and fermentation process was determined.

Although, the results are interesting, some details should be improved and explained.

I would like to make some comments that authors could take into account to improve the overall quality of the manuscript.

Comments:

The huge disadvantage is that the experimental results were not compared with control sample, i.e. dough with typical concentration of sodium chloride.

Authors should also comment which rate of sea salt and dry sourdough might give the best results as substitute of NaCl.

I have not noticed a test baking based on a new recipe and its sensory evaluation compared to the control sample.

Table 2 and Table 3: I do not understand why some non-significant effects (constants) were included into models whereas some effects were eliminated. The models with R2 lower than 70% should not be presented in manuscript; some of these models explain less than 50% of total variance (they are not predictive despite of significance of model, i.e. p < 0.05).

Fig. 2 and Fig. 3. It is much better if response surfaces are semi transparent. It gives opportunity to evaluate how well model was fitted (to see the residuals).

When discussing the results should be emphasized, what's new in this work, and not limit discussion into statements that similar results have been presented in the literature.

Author Response

Reviewer 1 comments:

Title: The effect of sodium reduction by sea salt and dry sourdough addition on the wheat flour dough rheological properties.

The effect of sea salt and dry sourdough from wheat flour as substitutes for sodium chloride on dough rheological properties on mixing, extension, pasting and fermentation process was determined.

Although, the results are interesting, some details should be improved and explained.

I would like to make some comments that authors could take into account to improve the overall quality of the manuscript.

Response: We would like to thank to the referee for its close reading of our manuscript. We hope that we improved the overall quality of the manuscript in especially by the addition of the sub points 3.6 and 3.7 in the manuscript.

Comments: The huge disadvantage is that the experimental results were not compared with control sample, i.e. dough with typical concentration of sodium chloride.

Response: It is an interesting point of view of the referee. Fortunately for us, we had these determinations made, determinations that we have wanted to be the subject of another article. However, in order to comply with the referee request we added this data in the manuscript. The data are presented now to the sub point 3.7. Also, within this sub point we made a comparison with the data obtained by us through optimization. So, according to the referee suggestion we compared now our experimental results with control sample, namely dough with typical concentration of sodium chloride.  

Authors should also comment which rate of sea salt and dry sourdough might give the best results as substitute of NaCl.

Response: We agree with the referee point of view. One of the main propose of the manuscript is to offer the best results as substitute of NaCl. So, we underlined these aspects in a quite large extensive way in our manuscript even in the first article version. In the sub point 3.5
was discussed only these aspects, namely comments on which rate of sea salt and dry sourdough might give the best results as substitute of NaCl. More, we have now been added and the sub points 3.6 and 3.7 where the optimum formulation of SS and SD is also disused through a comparison with control sample and also by framing in a legislating form through its sodium content.

I have not noticed a test baking based on a new recipe and its sensory evaluation compared to the control sample. Response: We would like to thank to the referee for her/his recommendation. Unfortunately, we did not have yet these determinations. Due to the COVID problem we cannot perform these tests now. Our city is in quarantine and we are not allowed to go to the laboratory to make them.  However, we want to focus on this manuscript on dough rheological properties only (technological aspects very important to the bakery producers). The baking test we want to be the subject of a future manuscript.

Table 2 and Table 3: I do not understand why some non-significant effects (constants) were included into models whereas some effects were eliminated. The models with R2 lower than 70% should not be presented in manuscript; some of these models explain less than 50% of total variance (they are not predictive despite of significance of model, i.e. p < 0.05).

Response: The constants are part of the mathematical model provided by the statistical processing method. They cannot be eliminated. The mathematical models that does not contain terms for some effects (such as A•B, A2or B2) are linear or 2FI models type. In order to comply to the referee request, in the Table 2 and Table 3 the parameters with R2 less than 70% (DT, FN, Tg and VR) were removed from tables.

Fig. 2 and Fig. 3. It is much better if response surfaces are semi transparent. It gives opportunity to evaluate how well model was fitted (to see the residuals).

  Response: In order to comply with the referee point of view we changed the response surface aspects from the entire manuscript (Fig 2 and Fig.3) according to his/her suggestions.

When discussing the results should be emphasized, what's new in this work, and not limit discussion into statements that similar results have been presented in the literature.

Response: We underline in the manuscript that the combination between sea salt and dry sourdough is for the first time discussed by us in the literature. We discussed in the manuscript only similar researches who discussed the individual effect of SS or SD. The combined effect of them is novelty information and we underlined that. Also, in many parts of the manuscript we underlined that this behavior may be due…..  so it is a supposition and not a certain fact. Also, the effect of SS on dough rheology was very less studied. Due to the fact that SS is a combination between magnesium chloride, potassium chloride, sodium chloride we tried to explain its behavior based on previously researchers made with individual chloride salts on dough rheology. So we do not agree with the referee point of view that our discussion is limited to the similar results. We think that we extended the discussion and explanations to explain novelty aspects such as SS and SD impact on dough rheology.   We look forward to hear from you soon. Sincerely yours,Mr. Silviu-Gabriel STROE et co.

Reviewer 2 Report

The manuscript by Voinea and and collaborators investigate a technological approach to decrease the sodium content from bakery products in order to respond to WHO recommendation to reduce the dietary salt intake. The study is very interesting and can contribute to reach WHO reccomendations. The manuscript is clear, well-written and reasonably discussed but I think that lack a deeper comparison with the amount of salt present in common bread and the ones proposed in the experiments. I mean that I'd like to see a discussion that shows a real decrease of salt (Na) when compared to a "standard bread".

Moreover there are few minor corrections which need attention such as:

line 44 and 45 are non clear, please rephrase

line 72 United States in brackets

line 100 CO2 instead of CO2

Author Response

Reviewer 2 comments:

The manuscript by Voinea and collaborators investigate a technological approach to decrease the sodium content from bakery products in order to respond to WHO recommendation to reduce the dietary salt intake. The study is very interesting and can contribute to reach WHO recommendations. The manuscript is clear, well-written and reasonably discussed but I think that lack a deeper comparison with the amount of salt present in common bread and the ones proposed in the experiments. I mean that I'd like to see a discussion that shows a real decrease of salt (Na) when compared to a "standard bread".

Response: We would like to thank to the referee for its appreciation and it’s the close reading of our manuscript. In order to comply with she/he suggestions we added now in the manuscript the sub point 3.6 in which we made a real discussion which shows the real decrease of salt (Na) when compared to a "standard bread".

Moreover there are few minor corrections which need attention such as:

line 44 and 45 are non clear, please rephrase

Response: We would like to thank to the referee for the close reading of the manuscript. We rephrased it.

line 72 United States in brackets

Response: We put it in the brackets now.

line 100 CO2 instead of CO2

Response: We made the corrections.  We look forward to hear from you soon. Sincerely yours,Mr. Silviu-Gabriel STROE et co.

Round 2

Reviewer 1 Report

Title of the manuscript is suitable. Abstract is enough informative. Materials and methods are suitable and well described. Discussion is based on the results. The paper gives to reader interesting and relevant information.

The paper has been corrected with accordance to my comments and I think that final version of this paper can be considered for publication.